# Personalized Physical Activity Programs for the Management of Knee Osteoarthritis in Individuals with Obesity: A Patient-Centered Approach

**DOI:** 10.3390/diseases11040182

**Published:** 2023-12-14

**Authors:** Hassan Zmerly, Chiara Milanese, Marwan El Ghoch, Leila Itani, Hana Tannir, Dima Kreidieh, Volkan Yumuk, Massimo Pellegrini

**Affiliations:** 1Orthopaedics and Traumatology Unit, Villa Erbosa Hospital, 40129 Bologna, Italy; zmerly@msn.com; 2Ludes Campus, 6912 Lugano, Switzerland; 3Department of Neurosciences, Biomedicine and Movement Sciences, University of Verona, 37131 Verona, Italy; chiara.milanese@univr.it; 4Center for the Study of Metabolism, Body Composition and Lifestyle, Department of Biomedical, Metabolic and Neural Sciences, University of Modena and Reggio Emilia, 41125 Modena, Italy; massimop@unimore.it; 5Department of Nutrition and Dietetics, Faculty of Health Sciences, Beirut Arab University, Riad El Solh, Beirut P.O. Box 11-5020, Lebanon; l.itani@bau.edu.lb (L.I.); h.tannir@bau.edu.lb (H.T.); d.kraydeyeh@bau.edu.lb (D.K.); 6Division of Endocrinology, Metabolism and Diabetes, Cerrahpaşa Medical Faculty, Istanbul University-Cerrahpaşa, 34363 Istanbul, Türkiye; vdyumuk@istanbul.edu.tr

**Keywords:** adherence, exercise, knee osteoarthritis, non-surgical treatment, obesity, barriers, pain, physical activity, physical fitness

## Abstract

Physical activity (PA) plays a vital role in knee osteoarthritis (KOA) management. However, engaging individuals with KOA in regular exercise is challenging, especially when they are affected by obesity. The aim of the current review is to elucidate how to increase adherence to exercise in this population. When implementing a PA program with patients with KOA and obesity, a specific multi-step approach can be adopted. In phase I (the baseline assessment), the patients’ eligibility for exercise is ascertained and a physical fitness assessment, sarcopenic obesity screening and quantification of the pain experienced are undertaken. Phase II adopts a patient-centered approach in implementing a PA program that combines an active lifestyle (>6000 steps/day) with land- or water-based exercise programs performed over eight to twelve weeks, with a frequency of three to five sessions per week, each lasting 60 min. In phase III, several strategies can be used to increase the patients’ adherence to higher levels of PA, including the following: (i) personalizing PA goal-setting and real-time monitoring; (ii) enhancing physical fitness and the management of sarcopenic obesity; (iii) building a sustainable environment and a supportive social network for an active lifestyle; and (iv) reducing pain, which can ameliorate the clinical severity of KOA and help with weight management in this population.

## 1. Introduction

Knee osteoarthritis (KOA) is a common progressive multifactorial joint disease that causes disability [1]. Recently, the global prevalence of KOA was estimated to be approximately 25% in middle-aged and older adults, and it has a significant impact on health, workplace productivity and economic costs [2]. Obesity is another clinical condition affecting public health that is also known to be associated with several comorbidities, including type 2 diabetes, cardiovascular diseases and some forms of cancer, which lead to an increase in disability and mortality [3]. According to the World Health Organization, in 2022, there were approximately 650 million adults globally with obesity, and the World Obesity Federation estimates that by 2030, one in five women and one in seven men will be affected by obesity [4]. Finally, obesity has been shown to be the most significant modifiable risk factor for the development and progression of KOA [5].

Physical activity (PA) is defined as any voluntary body movement by skeletal muscles that involves energy expenditure and can be categorized into occupational, sports, conditioning, household and other types [6]. The role of regular PA in the prevention and treatment of several chronic diseases has been widely reported [7]. In this context, recent evidence has underlined the importance of regular PA in patients with KOA since it appears to be associated with improved physical function and health-related quality of life (HRQoL) over time and has therefore been considered clinically useful in KOA management [8]. Similarly, PA appears to have a fundamental role in obesity management [9] for different reasons: first, because of its contribution to the preservation of muscle mass loss during weight loss from calorie restriction [10]; second, in being a predictive factor associated with better long-term weight loss maintenance [11]; and third, for its benefits in several health outcomes beyond reducing body weight, including a positive impact on conditions such as osteoarthritis (OA) [12]. However, engaging patients with KOA in regular PA remains challenging [13], especially among those who are also with obesity.

The purpose of this narrative review [14] is to describe a practical approach, based on the available recent literature, for patients with KOA and obesity who are willing to initiate a multi-step exercise program (Figure 1) comprising the following: (i) an initial baseline assessment; (ii) engagement in a personalized PA program via a patient-centered approach; and (iii) the introduction of several novel strategies that can help patients with KOA and obesity to improve their adherence to exercise.

## 2. Materials and Methods

This review was conducted following the guidelines of the Academy of Nutrition and Dietetics for narrative reviews, and the relevant seven-item checklist was completed (Appendix A) [15]. We searched the PubMed/Medical Literature Analysis and Retrieval System Online (MEDLINE) database for relevant literature related to KOA and obesity, and to PA and exercise, using the medical subject headings (MeSH) and their combinations: #1 = obesity, #2 = exercise, #3 = physical activity, #4 = physical fitness, #5 = physical performance, #6 = knee, #7 = osteoarthritis, #8 = cartilage damage, #9 = the Western Ontario and McMaster Universities Arthritis (WOMAC) index, #10 = knee osteoarthritis, #11 = pain, #12 = stiffness, #13 = clinical outcome, #14 = barriers and #15 = non-surgical. In addition, a manual search was conducted to find articles that were not identified in the initial strategy.

Papers on this topic were included if they met the following criteria: (i) published in the English language; (ii) regular investigation articles; (iii) prospective or retrospective observational, experimental or quasi-experimental studies, or systematic reviews or meta-analyses. Non-original papers (i.e., editorials and letters to the editor) were excluded. No limitations on the date of the publications were imposed, but the main focus was on papers published since 2015 in order to report updated findings.

## 3. Results

### 3.1. Phase I: Initial Assessment

A baseline assessment is conducted, evaluating the four aspects reported below and in Figure 2.

*Diagnosis of KOA and Obesity*—After considering the patient’s medical history and performing an accurate physical examination, X-ray and ultrasound, a diagnosis of KOA should be confirmed or rejected [16] via the Kellgren and Lawrence classification system, as detailed in Table 1 [17]. Screening for obesity and determining its clinical severity through anthropometric and body composition measures should then occur, as reported below and in Table 1.
○Body mass index (BMI) is the ratio of body weight in kg to height squared in meters (kg/m^2^). It is simple to obtain since it does not require sophisticated measurement tools, which means it is widely used in clinical settings to determine weight-related risk factors [18]. It can be determined according to the standard formula of body weight in kg divided by height in meters squared, measured with calibrated scales and a stadiometer, respectively. The patient is usually assessed while wearing lightweight clothing and no shoes. In Caucasians, a BMI ≥30 kg/m^2^ is normally indicative of obesity, categorized into three different classes: class I: BMI 30–34.9 kg/ m^2^; class II: BMI 35–39.9 kg/m^2^; class III: BMI ≥ 40 kg/m^2^, or specific cut-off points based on age, gender and ethnicity [19,20] (Table 1).○Waist circumference (WC) is a measurement of abdominal adiposity in centimeters, obtained via a tape measure at the level of the iliac crest while the individual is breathing normally. In Caucasians, a WC > 88 cm in females or 102 cm in males (in Europids, 80 cm in females, 94 cm in males) usually indicates abdominal obesity [21]. There are also age, ethnicity and gender-specific cut-offs that have been suggested (Table 1).○Body fat percentage (BF%) is defined as the amount of fat in the body, expressed frequently in kg. To determine the BF%, different techniques are used, e.g., bioelectrical impedance analysis (BIA), dual-energy X-ray absorptiometry (DXA), hydrostatic underwater weighing, air displacement plethysmography (BOD-POD), computed tomography (CT) scanning or magnetic resonance imaging (MRI) [22]. Age, ethnicity and gender-specific cut-offs have been suggested [23] (Table 1). 

By the end of this step, patients are categorized according to the clinical severity of KOA and obesity (e.g., stage II/class I; stage IV/class III, etc.).

*Eligibility for Exercise*—It is recommended that patients with KOA and obesity undergo medical evaluation for exercise eligibility in order to exclude the presence of other chronic diseases or a family history of cardiovascular diseases that may contraindicate exercising. Patients who reply with ‘no’ to any item reported in Table 2 can start a moderate-intensity PA program. Patients who report an affirmative response to at least one item should undergo a specialist second-level evaluation based on the chronic diseases reported in a checklist to determine their eligibility to exercise [24] (Figure 2 and Table 2).

*Assessment of Physical Fitness and Performance*—Physical fitness and performance are the ability of an individual to execute daily activities and are based on several components: body composition, cardiorespiratory endurance, flexibility and muscular strength and endurance [25]. It is vital to assess physical fitness and performance in order to understand the capacity and ability of the patient to exercise. Such an assessment also supports the development of a personalized PA program of a suitable nature and duration, based on a set of physical function performance tests for use in people diagnosed with KOA, as recommended by an international, multidisciplinary expert advisory group and endorsed by the Osteoarthritis Research Society International (OARSI). These include a 30 s chair stand test (30 s-CST), a 40 m fast-paced walk test (40 m FPWT), a stair-climb test, a timed up and go test (TUG) and a six-minute walk test (6 MWT), with normative values as indicated for people diagnosed with KOA [26].*Screening for Sarcopenic Obesity (SO)*—SO is defined as an increased body fat deposition and decreased muscle mass and strength [27]. Recent reports have demonstrated that SO seems to negatively impact therapeutic and surgical outcomes in patients with KOA [28]; hence, it is vital to screen for SO in this population [29]. However, to the best of our knowledge, very few physical performance tests and tools are available to screen for SO in patients with KOA. Recent work has shown that hand grip strength adjusted by BMI, with cut-offs below 0.65 in females and 1.1 in males, can be indicative of a higher risk of SO in people with KOA [30]. A more accurate assessment should be conducted in order to confirm the diagnosis of SO. Appendicular lean mass (ALM), adjusted by body size (i.e., body weight (kg) or BMI (kg/m^2^) seem to be clinically reasonable in this population. Specifically, ALM/BMI cut-off points of <0.512 in females and <0.789 in males appear to be suitable for the identification of SO in a clinical population with KOA [30].*Quantification of the Pain Experience*—Pain is a highly unpleasant physical sensation caused by illness or injury [31]. It is affected by physiological, psychological and demographic factors, which cause great variation in individual perception. The experience of pain seems to affect the participation of individuals with KOA in PA [32]; therefore, quantifying the pain experience in this population is an important first step before its management. Several self-report tools are available for this purpose, such as the short-form McGill Pain Questionnaire (MPQ-SF) [33].

### 3.2. Phase II: Patient-Centered Approach: Engagement in a Personalized Physical Activity Program


**Patient-Centered Communication**


In phase II, two important aspects need to be assessed and evaluated: first, the patient’s knowledge about the benefits of exercise for both KOA and obesity; second, their motivation to exercise (Figure 1 and Figure 3). This can be discussed openly and/or measured via a validated questionnaire that generally assesses the individual’s knowledge about the positive impact of physical exercise on health. Motivation for PA can be assessed through validated questionnaires (i.e., the motivation for physical activity questionnaire, RM4-FM) [34]. In both cases, the clinician should not adopt a judgmental attitude towards the information the patient reports on the topic.

Patients should be actively involved in the decision to initiate PA and are invited to think about the reasons for and against exercising [35]. It is best to start by asking patients to list the disadvantages of changing their lifestyle, such as whether a sedentary life conveys perceived advantages that they are afraid of losing or unwilling to lose [35]. Subsequently, patients should be asked to evaluate the advantages; clinicians should encourage them to reflect on both the short- and long-term effects of exercise on body weight control and on the clinical severity of KOA [35]. The list of advantages and disadvantages should be written down in a table form for future reference for discussion [35]. This table should then be analyzed in detail together with the patient. During this discussion, the clinician should help patients not only focus on the immediate future but also on their long-term goals [35]. Every reason for change should be reinforced. It is also important to address the disadvantages of adjusting lifestyle, with the aim being to help patients to conclude themselves that exercise is necessary to control body weight in the long term and to improve the clinical severity of KOA [35].

Based on their preferences, motivations, needs and abilities, the PA program should be personalized rather than generic. Evidence shows that adherence to exercise increases when it is less structured and that patients are more likely to engage in PA when instructed to do so on their own at home than when asked to attend supervised, group-based, on-site exercise sessions [36]. Finally, multiple short bouts of exercise (10 min each), rather than one long session, may help patients accumulate more daily minutes of exercise by providing them with more easily achievable goals [37]. There are two types of physical exercise.


**Patient-Centered Engagement in a Personalized Exercise Program**
○*Active Lifestyle (Expressed as Steps/Day)*—The primary aim of developing an active lifestyle is to reduce the time spent in sedentary behaviors by working on the PAs that are part of everyday life (e.g., walking, standing, climbing stairs, cleaning the house, gardening, etc.). It is best expressed as the number of steps per day and is easily monitored via simple tools such as pedometers [38]. An early observational longitudinal study conducted in an elderly population with obesity revealed that walking is associated with a lower risk of functional limitations [39]. A cut-off of 6000 steps/day appeared to be protective, and each additional 1000 steps/day was associated with an approximately 20% reduction in the risk of functional limitation [39]. In another, recently published study undertaken with patients with KOA who were overweight/obese, participants who did not report regular knee pain and who regularly walked for exercise were less likely to later develop knee pain (26%) at a follow-up eight years later compared with those who did not (37%) [40] (Table 3). The importance of an active lifestyle (expressed in steps/day) stems from its impact on weight control. Early data derived from the U.S. National Weight Control Registry showed that people with successful long-term weight-loss maintenance share common behaviors, such as adhering to a low-fat diet, engaging in frequent self-monitoring of body weight and food intake, and having high levels of regular exercise, specifically 10,000 steps/day [41]. More recent work has shown that walking 11,000 steps or more prevents overweight individuals from developing obesity after four years of follow-up by nearly 64% [42]. An active lifestyle should be prioritized in this population (i.e., those with KOA who are affected by overweight/obesity) because of its dual impact.

**Formal Exercise**
○*Land-Based Exercise*—The European Society for Clinical and Economic Aspects of Osteoporosis, Osteoarthritis and Musculoskeletal Diseases (ESCEO) [43] and the OA Research Society International (OARSI) [44,45] proposed strong recommendations for land-based exercises for the non-surgical management of KOA. These recommendations were based on high-quality evidence derived from several systematic reviews and meta-analyses of randomized controlled trials (RCTs) that identified significant clinical benefits in the short term (two to six months), especially in relation to pain reduction and physical function improvement [46,47,48,49,50]. The nature (i.e., duration and type of exercise) of the programs in these guidelines varied widely from strength training to a range of motion exercises and aerobic activity [46,47,48,49,50]. More recent work has specifically recommended activities such as Pilates and aerobic and strengthening exercise programs, which have been shown to have beneficial effects, especially on pain and strength, when performed over eight to twelve weeks for a total of three to five sessions per week. Each session must last at least 1 h to be effective [51] (Table 3).○*Water-Based Exercise*—The hydrostatic pressure and resistance of water, especially in warm temperatures (i.e., heated pools), create a beneficial environment for patients with KOA [52]. Specifically, in water, the knee is less overloaded by body weight, the muscles supporting the knee can be strengthened in a non-traumatic way and the blood flows better [52]. Therefore, water-based exercise is recommended for patients with severe KOA and obesity, or for those with a proprioceptive deficit [52]. An earlier systematic review of RCTs assessing the effectiveness of water-based exercise on KOA found that this approach results in improvements in self-reported pain and disability, but that the effect seems to be of a small magnitude and short duration [53]. Recently, another systematic review and meta-analysis with more specific recommendations was conducted [54] based on RCTs involving administered water-based exercise programs in pools in which the water was between waist and chest height and at a temperature of 28–34 °C, with a duration/session of 50 to 60 min for two to five sessions/week, and over an intervention program period of 12 weeks. These interventions were found to alleviate pain, improve the quality of life and reduce dysfunction [54] (Table 3). However, more research is still needed to clarify the exact type of such water-based exercise.


### 3.3. Phase III: Strategies to Increase Adherence to Exercise

*Education on the Benefits of PA for Health, and Motivating Patients to Exercise*—In general, education is a key factor in OA management [55]. People with greater knowledge about the benefits of PA for health tend to be more active [56]. It is therefore useful to highlight the benefits of regular PA for both KOA and obesity and discuss with patients the available evidence regarding the impact of PA (Figure 3) in (i) decreasing pain, (ii) improving function and (iii) enhancing HRQoL [51]. PA has several positive effects on obesity [12] that are also worth highlighting to patients, such as (i) increased energy expenditure and enhanced adherence to caloric restriction [57]; (ii) preservation of muscle mass during weight loss [10]; (iii) maintaining long-term weight loss [58]; and (iv) enhancing body image and psychological outlook [59] (Figure 3). Moreover, the adoption of an engaging style is vital for motivating patients to exercise. Health professionals should, first of all, show empathy for the patient’s difficulties in exercising because of obesity and KOA, and always propose an achievable PA program for using a collaborative style [60] (Figure 4).

*Increase the Levels of Physical Fitness and Performance*—To improve cardiovascular fitness [61], aerobic outdoor (i.e., walking, cycling, swimming, jogging, etc.) and/or indoor (i.e., exercise bike, climbing stairs, aerobic gymnastics, etc.) activities, when undertaken correctly at a certain intensity, duration and frequency, may have numerous benefits, such as enhancing cardiorespiratory fitness [62]. Calisthenic gymnastics [63], which consist of the use of body weight and gravity, can be used to increase muscular tone and strength via continuous repetitions of a certain exercise, which is usually called a ‘set’. The increase in the number of sets of a certain exercise may boost the resistance and strength of a specific group of muscles. Muscular flexibility and elasticity can be developed through stretching exercises that consist of a unique slow stretching movement to a position of mild muscular tension but not pain, holding for 20 s. This process should be repeated two to three times with an increased level of muscular tension (Figure 1 and Figure 4).*Sarcopenic Obesity Management*—The management of SO in patients with KOA and obesity is just as important as the improvement of physical fitness. There are several recent publications relating to this specific population, covering different dietary and nutritional strategies that can potentially ameliorate the severity of KOA and, simultaneously, improve SO indices [64]. Adherence to a low-calorie Mediterranean diet can determine significant weight loss and remains the cornerstone nutritional approach in this population [64]. Supplements such as vitamin D, essential and non-essential amino acids and whey protein also appear to be beneficial to both KOA and SO [64].*Pain Management*—Pain is a barrier to PA that is consistently reported by people with KOA. Its management remains an important target of interventions, especially in its acute phase and involves a range of strategies, including the following: (i) over-the-counter medications (i.e., paracetamol) [65]; (ii) oral non-steroidal anti-inflammatory (NSAIDs) (i.e., cyclooxygenase (COX) inhibitors), corticosteroids, opioids (i.e., tramadol) and others (i.e., duloxetine, capsaicin) [65]; (iii) topical and intra-articular drugs (i.e., NSAIDs, corticosteroids, hyaluronic acid) [65]; (iv) acupuncture [66]; (v) physical therapy [67]; (vi) weight loss; and (vii) cognitive behavioral therapy [68].*Sustainable Environment and Social Support for a More Active Lifestyle*—Treatment should aim to modify the environment from a stigmatizing to a stimulating one [69] that supports changes towards instilling habits regarding exercise, free of architectonic barriers [70,71]. Patients are encouraged to reduce triggers for physical inactivity and increase positive cues for healthy PA [72]. Several studies suggest that social support is a key factor in behavioral changes and is considered to be an important aid for weight loss maintenance [73]. Specifically, more social support, especially from relatives, is associated with higher levels of leisure PA [74]. Therefore, it is important to involve patients’ families to create the optimum environment for change, since they can be crucial in encouraging patients to change and increase their level of PA [55]. It is vital that significant others are educated about obesity, KOA and physical exercise, and encouraged to be actively involved in exploring how to help patients develop and maintain an active lifestyle [55]. Although it is important to consider that needs vary from patient to patient, the general advice to give to family and friends includes creating a relaxed environment, reinforcing positive behaviors, adopting a positive attitude, exercising together and accepting patients’ difficulties, as ambivalence can lead to setbacks [55].*Personalized Goal-Setting and Real-Time Monitoring*—Patients are encouraged to set specific and quantifiable weekly PA goals that are challenging but realistically achievable [75]. Reaching goals leads to self-reinforcement and self-efficacy enhancement [76]. Patients should start with gentle exercising and gradually increase to a weekly goal, as per the PA recommendations in primary care [77]. For instance, walking (i.e., an active lifestyle) is the preferred PA for patients with KOA [39,40,78] since it is a form of unstructured PA that can be easily implemented in a daily routine, with a goal to increase daily steps by 500 a day to reach 10,000 daily steps. If patients are willing to initiate a formal form of land- or water-based PA, this is also advisable, and strategies such as encouraging patients to exercise with a family member or friend, enroll in a club or gym or seek help from a personal trainer are all good methods that can help them to increase their adherence to this type of PA. In both cases (daily steps or formal PA), the setting of unrealistic goals should be promptly discouraged when discussing this topic with the patient. Finally, in line with personalized goal setting for PA, the self-monitoring of PA, whether through structured or unstructured forms, is another strategy in which real-time monitoring raises patients’ awareness of their exercise habits and helps them to improve and increase their levels of PA [79]. To achieve this aim, PA can be recorded on a monitoring record in minutes (of programmed activity) and/or number of steps (of lifestyle activity) using a pedometer [79].

## 4. Discussion

### 4.1. Findings

This narrative review has attempted to describe a novel approach to how a PA program can be implemented in the management of patients with comorbid KOA and obesity. This population is known to face significant obstacles in adhering to PA through an active lifestyle and/or a structured PA. This review could be used as a guide by various health professionals (i.e., orthopedists, rheumatologists, nutritionists, dieticians, obesity specialists, physical therapists, etc.) who usually work with such patients. The originality of this review lies in its format as a non-prescriptive paper, which means it can go beyond making general, technical recommendations. The review has articulated a three-phase approach. The phase I assessment stage considers the patient’s clinical condition in a personalized way (i.e., clinical severity, eligibility to exercise, physical fitness, etc.). Phase II involves the tailored implementation of a PA program via a patient-centered approach (i.e., individual needs and capacities). In phase III, common obstacles to exercise are identified, and patients are provided with novel strategies to overcome them and consistently adhere to higher levels of PA.

### 4.2. Clinical Implications

The findings of our narrative review may have several clinical implications. First, when dealing with patients with KOA and obesity, clinicians should be aware of and highlight to their patients the benefits of adhering to an active lifestyle and/or a structured PA program for the management of KOA and obesity. At the same time, they should openly discuss the challenges and barriers that the patients will experience. Second, clinicians should instill hope in patients that there are effective strategies to help them overcome these barriers, increase their adherence to exercise and improve their clinical outcomes (i.e., KOA clinical severity and obesity status).

### 4.3. Strengths and Limitations

To the best of our knowledge, this narrative review is the first to exclusively consider a specific population of individuals with KOA and obesity and to propose an approach that can improve adherence to PA in order to ameliorate the clinical severity of KOA and help in the management of obesity. The majority of findings in this review, especially those related to phase II (PA program implementation via formal land- or water-based exercises or via an active lifestyle) derive from solid evidence (i.e., ESCEO and OARSI guidelines, multiple systematic reviews, RCTs and longitudinal observational multicenter studies).

There are also some limitations. First, the review is narrative and not a systematic review or meta-analysis [80]; however, it may be premature to conduct a systematic review because of the heterogeneity of the available literature or/and the few studies on this specific population with comorbid obesity and KOA. Second, our findings should be interpreted with caution since some included studies have small samples or use a cross-sectional design. Replications in large longitudinal studies would be useful to determine a cause–effect of PA on both obesity and KOA. Finally, our findings cannot be generalized to all patients with KOA and obesity, e.g., those with severe clinical obesity who are candidates for surgery [81].

### 4.4. New Directions for Future Research

There is an urgent need to direct future investigations towards other new strategies that can improve and increase the adherence of patients with KOA and obesity to an active lifestyle, exercise and structured PA programs. Similar reviews should be conducted to better describe the implementation of PA and exercise in the same population with major clinical severity (grades 3 and 4), or in those patients who have undergone surgery (i.e., knee arthroplasty) [82].

## 5. Conclusions

Obesity and KOA are two prevalent chronic diseases that have negative impacts on individual patients’ physical wellbeing and represent a burden on public health systems [83,84,85]. They often co-exist [5], resulting in the experience of major pain (caused by a greater inflammatory status), more impaired physical mobility, additional limitations in functional activities [86,87] and an increase in physical inactivity [88]. This becomes a vicious cycle that leads to clinical deterioration in both conditions. PA plays an important role in the management of KOA and obesity, but it remains an unfamiliar construct for many patients and for healthcare professionals. Low adherence to PA and the attrition of the patient from PA interventions remain areas of great concern yet have received little attention. The present review represents a practical guide that describes the implementation of personalized PA programs for patients with KOA and obesity, and a series of strategies that can increase adherence to these programs.

## Figures and Tables

**Figure 1 diseases-11-00182-f001:**
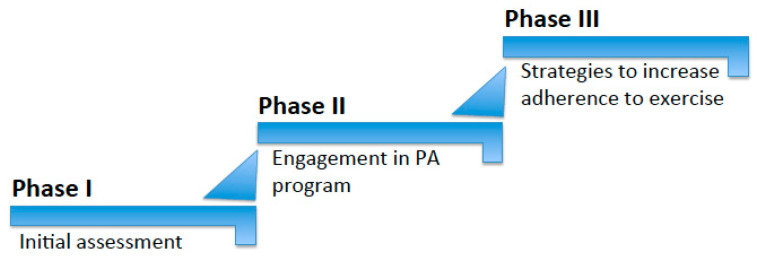
Multi-step exercise program for individuals with KOA and obesity.

**Figure 2 diseases-11-00182-f002:**
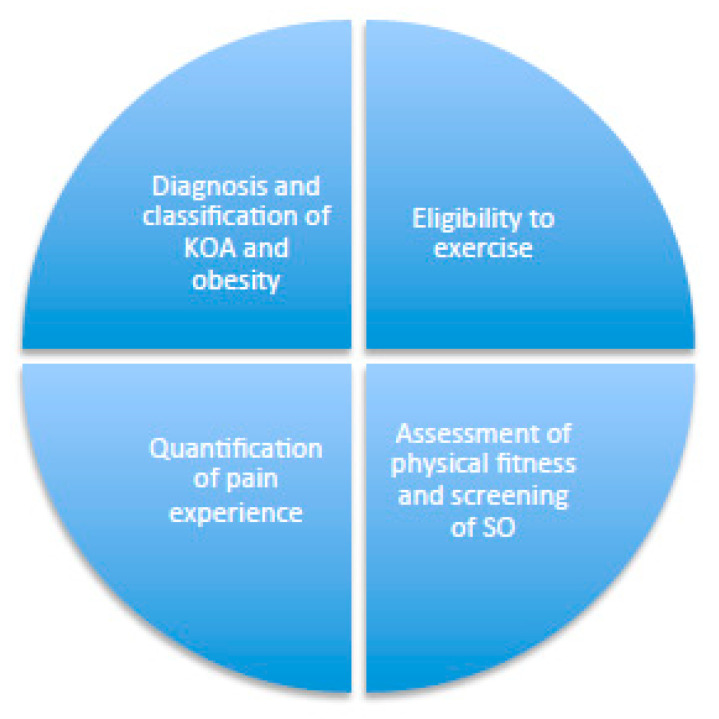
Phase I: initial assessment.

**Figure 3 diseases-11-00182-f003:**
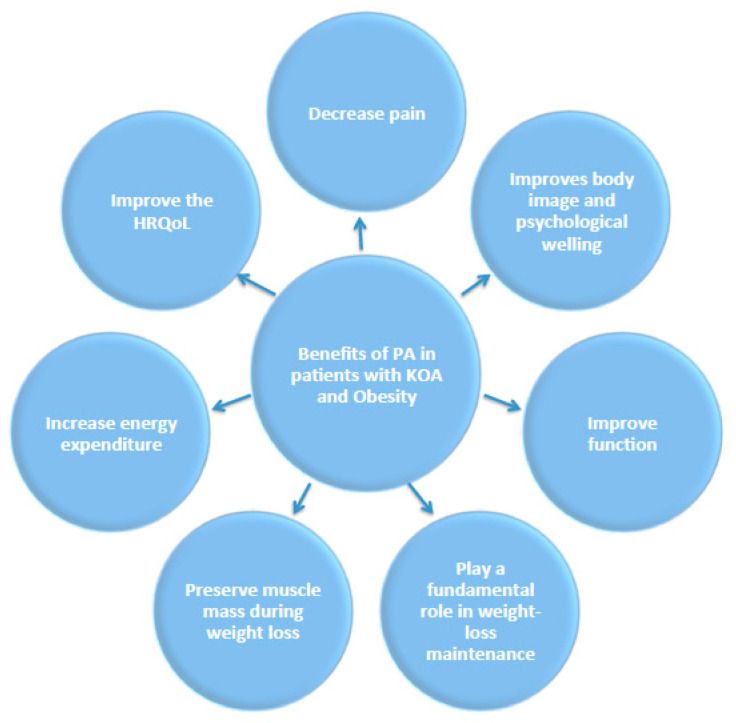
The beneficial impact of regular exercise in patients with KOA and obesity.

**Figure 4 diseases-11-00182-f004:**
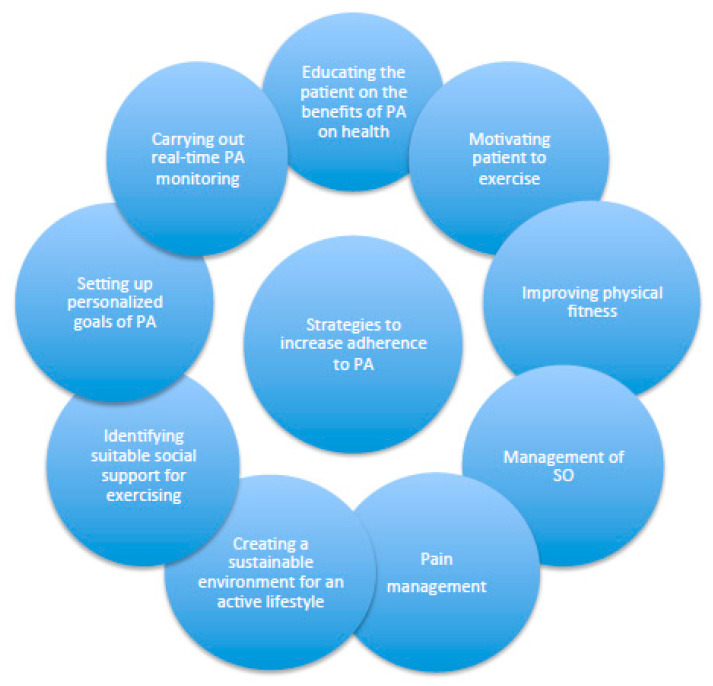
Phase III: strategies to increase adherence to higher levels of PA in patients with KOA and obesity.

**Table 1 diseases-11-00182-t001:** Diagnostic criteria for obesity and KOA classification according to the Kellgren and Lawrence classification system.

Obesity			
Tool		Cut-Off Points	
		Females	Males
BMI (kg/m^2^) *		≥25–30 kg/m^2^	≥25–30 kg/m^2^
WC (cm) *		>80–88 cm	90–102 cm
BF (%) *		38–43%	26–31%
* Age, ethnicity and gender-specific cut-off points			
**Kellgren and Lawrence KOA Classification System**			
Grade 0	No radiological findings of OA		
Grade I	Doubtful joint space narrowing and possible osteophytic lipping		
Grade II	Certain osteophytes and possible joint space narrowing		
Grade III	Moderate multiple osteophytes, certain narrowing of joint space, some sclerosis and possible deformity of bone ends		
Grade IV	Large osteophytes, marked narrowing of joint space, severe sclerosis and certain deformity of bone ends		

Abbreviations: BMI = body mass index; WC = waist circumference; BF% = body fat percentage; KOA = knee osteoarthritis; OA = osteoarthritis.

**Table 2 diseases-11-00182-t002:** Ten-item questionnaire to assess indications of chronic diseases during medical examinations.

Item	Yes	No
In the past, the patient suffered from one of the following diseases: heart infarction, angina pectoris, heart arrest, cardiac arrhythmia, cardiac valves disease		
Has blood hypertension		
Has been followed up by a cardiologist		
Had an ictus or any neurological problem		
Has type 1 or 2 diabetes		
Has endocrine metabolic diseases		
Has liver or renal diseases		
Has orthopedic or skeletal problems		
In the past, had any health problem considered a barrier to doing physical activity		
The patient thinks that physical activity may be harmful or risky for him/her		

**Table 3 diseases-11-00182-t003:** Phase II: personalized PA program.

	PA Program	Effect	Evidence
Active lifestyle (steps/day)	Approx. 6000 steps/day and additional 1000 steps/day Regular walking	Protect individuals with KOA and obesity from having functional limitations (6000 steps/day), and each additional 1000 steps/day may reduce the risk of functional limitation by nearly 20%. In individuals who are overweight/obese and have KOA/no pain, regular walkers are less likely to develop pain at an eight-year follow-up.	Longitudinal, multi-center observational study [39] Nested cohort study [40]
Land-based exercise (Time/session/week)	Pilates, aerobic and strengthening exercise program, at least 1 h/session, three to five sessions per week, performed for eight to 12 weeks	Clinical benefits in the short term (two to six months), especially for pain reduction and physical function improvement	Systematic review and meta-analysis [46,47,48,49,50,51]
Water-based exercise (Time/session/week)	Programs in pools, with water between waist and chest height at 28–34 °C, for a duration of 50 min per session, two to five sessions/week, performed for 12 weeks	Pain reduction and disability improvement of small magnitude and of short duration Alleviate pain, increase quality of life and reduce dysfunction.	Systematic review and meta-analysis [53] Systematic review and meta-analysis [54]

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
