# Peer review of "Personalized Physical Activity Programs for the Management of Knee Osteoarthritis in Individuals with Obesity: A Patient-Centered Approach"

_diseases, 2023, doi:10.3390/diseases11040182_

Round 1

Reviewer 1 Report

Comments and Suggestions for Authors

This article, "Personalized Physical Activity Programs for the Management of Knee Osteoarthritis in Individuals with Obesity: A Patient-Centered Approach," reviews the connections between physical activity (PA) and knee osteoarthritis (KOA). The manuscript is generally well-written; however, it can be improved in the following ways:

  1. In terms of figures, the article does not need excessive color if it does not aid in readability. For example, Figure 1 could simply be a step-by-step demonstration. In Figure 2, the text need not be vertical and could be oriented horizontally. Figure 4 does not require color either. A flattened design for the figures would enhance readability.
  2. The Materials and Methods section details a narrative review approach. While this is appropriate, it would benefit from a more explicit justification for choosing a narrative review over other review types, such as systematic reviews or meta-analyses. Additionally, specifying the criteria for including studies and describing how the quality of these studies was assessed would strengthen the methodological rigor.
  3. Discussing the limitations of the studies reviewed, such as sample size, study design, or potential biases, would provide a more balanced perspective.
  4. English editing by an native speaker would also enhance readability 
Comments on the Quality of English Language
  1. English editing by an native speaker would also enhance readability 

Author Response

Reviewer 1 #

This article, "Personalized Physical Activity Programs for the Management of Knee Osteoarthritis in Individuals with Obesity: A Patient-Centered Approach," reviews the connections between physical activity (PA) and knee osteoarthritis (KOA). The manuscript is generally well written; however, it can be improved in the following ways:

  1. In terms of figures, the article does not need excessive color if it does not aid in readability. For example, Figure 1 could simply be a step-by-step demonstration. In Figure 2, the text need not be vertical and could be oriented horizontally. Figure 4 does not require color either. A flattened design for the figures would enhance readability. Response: Figure 1 is a step-by-step demonstration. Figure 2 now is simplified and the text is in horizontal. All colors that are not necessary are now removed from the figures (Check figures)
  1. The Materials and Methods section details a narrative review approach. While this is appropriate, it would benefit from a more explicit justification for choosing a narrative review over other review types, such as systematic reviews or meta-analyses. Additionally, specifying the criteria for including studies and describing how the quality of these studies was assessed would strengthen the methodological rigor. Response: we added in the Materials and Methods section the quality assessment tool for narrative reviews as indicated by the guidelines of the Academy of Nutrition and Dietetics for narrative reviews (Page 2, paragraph 4). Moreover we added more explicit justification for choosing a narrative review over other review types, such as systematic reviews or meta-analyses in the Discussion section specifically among the limitations (Page 14, paragraph 2).
  1. Discussing the limitations of the studies reviewed, such as sample size, study design, or potential biases, would provide a more balanced perspective. Response: The suggested limitations have been added in the Discussion section (Page 14, paragraph 2).
  1. English editing by an native speaker would also enhance readability Response: English editing was performed by a professional service.

Reviewer 2 Report

Comments and Suggestions for Authors

Materials and Methods

In line 95, it would be beneficial to add the range of years during which the articles were considered.

Results

Lines 101 and 111: I recommend removing the reference to "Figure 1" from the text. Instead, consider using a descriptive phrase like "3.1. Phase I: Initial Assessment (at Baseline)" to indicate the stage of the phases.

Lines 102-109: The information presented here is repeated in Table 1. I suggest including this information solely within the table and removing it from the text.

Line 112: Consider describing both Table 1 and Figure 1 within the text. For instance, you could mention that Figure 1 illustrates the sub-phases of Phase 1, while Table 1 provides the diagnostic criteria for obesity and the KOA classification according to the Kellgren and Lawrence system.

Line 169: I recommend removing the reference to "Figure 1" from the text.

Lines 167-169: In reference to the potential results in Table 2, please specify which patients are eligible for physical exercise.

Line 184: It would be beneficial to remove the references to "Figure 1 and 2" from the text.

Line 206: Similarly, consider deleting the references to "Figure 1 and 2" from the text.

Line 214: Remove references to "Figure 1 and 2" from this section. The current phrasing may lead the reader to anticipate the display of the McGill Pain Questionnaire short-form in the mentioned figures.

Figure 3: Enhance the image quality and position it after the paragraph spanning lines 218-226.

Line 220: Suggest eliminating the reference to "Figure 1" from the text. Use descriptive subtitles, such as "3.2. Phase II: Patient-Centered Approach: Engagement in a Personalized Physical Activity Program," to indicate the stage.

Line 242: Reference #35 does not sufficiently support the claim that exercise is essential for long-term body weight control and the improvement of KOA clinical severity. Please replace it with a more suitable reference.

Line 269: Recommend removing the references to "Figure 1 and Table 3" from the text. The current wording may suggest that the statistics from reference 40 will be shown in these figures.

Line 280: Suggest removing the references to "Figure 1 and Table 3" from the text.

Line 297: Similarly, remove references to "Figure 1 and Table 3" from this section.

Line 315: Suggest deleting references to "Figure 1 and Table 3."

Line 338: Remove the reference to "Figure 1" from the text.

Line 399: Suggest removing the reference to "Figure 1."

Lines 408-409: Delete references to "Figure 1 and 4."

Line 430: Remove references to "Figure 1 and 4."

Line 454: Delete the reference to "Figure 4."

Figure 4: Enhance the image quality.

General Suggestion:

Consider incorporating figures within the text in a descriptive manner rather than placing them solely in parentheses (e.g., Figure X or Table X). This approach enhances clarity and accessibility for readers.

Comments on the Quality of English Language

Minor spelling issues

Author Response

Reviewer 2 #

Materials and Methods

In line 95, it would be beneficial to add the range of years during which the articles were considered. Response: now it has been added the range of years during which the articles were considered (Page 3, paragraph 2).

Results

Lines 101 and 111: I recommend removing the reference to "Figure 1" from the text. Instead, consider using a descriptive phrase like "3.1. Phase I: Initial Assessment (at Baseline)" to indicate the stage of the phases. Response: removed the reference to "Figure 1" as suggested, and added a descriptive statement instead (Page 3, paragraph 3).

Lines 102-109: The information presented here is repeated in Table 1. I suggest including this information solely within the table and removing it from the text. Response: now we removed the information from the text and leave it only in Table 1 (Page 4).

Line 112: Consider describing both Table 1 and Figure 1 within the text. For instance, you could mention that Figure 1 illustrates the sub-phases of Phase 1, while Table 1 provides the diagnostic criteria for obesity and the KOA classification according to the Kellgren and Lawrence system. Response: now we added the suggested statement by the reviewer (Page 3, paragraph 4).

Line 169: I recommend removing the reference to "Figure 1" from the text. Response: removed the reference to Figure 1 as suggested (Page 5, paragraph 2).

Lines 167-169: In reference to the potential results in Table 2, please specify which patients are eligible for physical exercise. Response: now we specified which patients are eligible to exercise (Page 5, paragraph 2).

Line 184: It would be beneficial to remove the references to "Figure 1 and 2" from the text. Response: we removed the references to "Figure 1 and 2" from the text as suggested (Page 5, paragraph 3).

Line 206: Similarly, consider deleting the references to "Figure 1 and 2" from the text. Response: we removed the references to "Figure 1 and 2" from the text as suggested (Page 6, paragraph 1).

Line 214: Remove references to "Figure 1 and 2" from this section. The current phrasing may lead the reader to anticipate the display of the McGill Pain Questionnaire short-form in the mentioned figures. Response: we removed the references to "Figure 1 and 2" from the text as suggested (Page 6, paragraph 2).

Figure 3: Enhance the image quality and position it after the paragraph spanning lines 218-226. Response: we improved the quality of Figure 3 and move it as suggested (Page 7).

Line 220: Suggest eliminating the reference to "Figure 1" from the text. Use descriptive subtitles, such as "3.2. Phase II: Patient-Centered Approach: Engagement in a Personalized Physical Activity Program," to indicate the stage. Response: we eliminated the reference to Figure 1 as suggested (Page 6, paragraph 3).

Line 242: Reference #35 does not sufficiently support the claim that exercise is essential for long-term body weight control and the improvement of KOA clinical severity. Please replace it with a more suitable reference. Response: we replaced the reference with more suitable one:

Jurado-Castro JM, Muñoz-López M, Ledesma AS, Ranchal-Sanchez A. Effectiveness of Exercise in Patients with Overweight or Obesity Suffering from Knee Osteoarthritis: A Systematic Review and Meta-Analysis. Int J Environ Res Public Health. 2022;19(17):10510.

Line 269: Recommend removing the references to "Figure 1 and Table 3" from the text. The current wording may suggest that the statistics from reference 40 will be shown in these figures. Response: we removed the references to "Figure 1 and Table 3" from the text as suggested (Page 8, paragraph 1).

Line 280: Suggest removing the references to "Figure 1 and Table 3" from the text. Response: we removed the references to "Figure 1 and Table 3" from the text as suggested (Page 8, paragraph 1).

Line 297: Similarly, remove references to "Figure 1 and Table 3" from this section. Response: we removed the references to "Figure 1 and Table 3" from the text as suggested (Page 8, paragraph 2).

Line 315: Suggest deleting references to "Figure 1 and Table 3." Response: we removed the references to "Figure 1 and Table 3" from the text as suggested (Page 9, paragraph 1).

Line 338: Remove the reference to "Figure 1" from the text. Response: we removed the reference to "Figure 1 from the text as suggested (Page 11, paragraph 1).

Line 399: Suggest removing the reference to "Figure 1." Response: removed as suggested (Page 12, paragraph 2).

Lines 408-409: Delete references to "Figure 1 and 4." Response: we removed the references to "Figure 1 and 4" from the text as suggested (Page 12, paragraph 3).

Line 430: Remove references to "Figure 1 and 4." Response: Response: we removed the references to "Figure 1 and 4" from the text as suggested (Page 12, paragraph 4).

Line 454: Delete the reference to "Figure 4." Response: we removed the reference to "Figure 4 from the text as suggested (Page 13, paragraph 1).

Figure 4: Enhance the image quality. Response: The quality of Figure 4 has been improved.

General Suggestion:

Consider incorporating figures within the text in a descriptive manner rather than placing them solely in parentheses (e.g., Figure X or Table X). This approach enhances clarity and accessibility for readers. Response: we thank the reviewer, now we adhered to the general suggestion.

Round 2

Reviewer 2 Report

Comments and Suggestions for Authors

The authors have effectively addressed all the issues raised in the previous review, significantly improving readability, referencing, clarity of figures, and providing in-depth information about the study population, as well as the materials and methods used in the development of this paper. Given these substantial improvements, I believe the paper could be accepted after a minor spelling review.

Comments on the Quality of English Language

The authors have effectively addressed all the issues raised in the previous review, significantly improving readability, referencing, clarity of figures, and providing in-depth information about the study population, as well as the materials and methods used in the development of this paper. Given these substantial improvements, I believe the paper could be accepted after a minor spelling review.